# A deeper look at carrier proteome effects for single-cell proteomics

Zilu Ye [1], Tanveer S. Batth[1], Patrick Rüther [1] & Jesper V. Olsen [1✉]

Multiplexing approaches using tandem mass tags with a carrier proteome to boost sensitivity have advanced single cell proteomics by mass spectrometry (SCoPE-MS). Here, we probe the carrier proteome effects in single cell proteomics with mixed species TMTpro-labeled samples. We demonstrate that carrier proteomes, while increasing overall identifications, dictate which proteins are identified. We show that quantitative precision and signal intensity are limited at high carrier levels, hindering the recognition of regulated proteins. Guidelines for optimized mass spectrometry acquisition parameters and best practices for fold-change or protein copy number-based comparisons are provided.

[1] Novo Nordisk Foundation Center for Protein Research, Faculty of Health and Medical Sciences, University of Copenhagen, Copenhagen, Denmark.
✉email: jesper.olsen@cpr.ku.dk

Mass spectrometry-based single-cell proteomics (SCP-MS) has recently seen significant developments[1–3]. To overcome analytical barriers such as insufficient peptide ion signals for MS identification and quantification, a multiplexing strategy based on labeling tryptic peptides from single cells with isobaric tandem mass tags (TMT) alongside a labeled carrier proteome to boost MS signal has been developed[4]. Several studies have recently emphasized the importance of increasing the number of ions sampled from the single-cell channels (SCCs) with a carrier proteome channel (CPC), concluding that the depth of peptide identification needs to be balanced against the accuracy of quantification[5–7]. In this study, we highlight additional crucial factors for performing SCP-MS experiments, these include: (1) proper selection of the carrier proteome; (2) unneglectable isotope impurities caused by the carrier channel; (3) balance between signal-to-noise ratio (SNR), collisional energy and resolution; (4) suitability of SNR and intensity for different data interpretation strategies.

## Results

**Experimental setup with a mixed-species sample mixture**. We modeled an SCP-MS experiment using TMTpro[8] 16plex labeling reagents, where channel 126 severed as the CPC, 127 C was left empty, and the last 14 channels represented SCCs at different ratios to the CPC (Fig. 1a). To achieve this, we constructed a mixed-species sample from *homo sapiens* (HeLa cells), *Saccharomyces cerevisiae* (Yeast) and *Escherichia coli* (E. coli), which were pooled at different known ratios in the SCCs, in order to elucidate the bidirectional effect on identification and quantification in SCP (Fig. 1a). We investigated the effects of CPC quantities and proteome types by designing different CPC constructs with one of three different carrier proteomes: Human only (H), E. coli and yeast mixed (EY), and all three species (HEY) mixed across a large range (14–434×) of CPC to SCC ratios (hereafter as carrier levels). Together with samples without any CPC (no carrier), these samples were analyzed by liquid chromatography tandem mass spectrometry (LC-MS) with different MS parameters (Fig. 1a, Supplementary Note 1). Loading amounts (50–200 pg) per SCC were equivalent to single-cell proteomes[5].

**Carrier proteomes dictate which proteins are identified**. We first tested how different carrier proteomes affect protein identifications. We compared the number of non-human and human proteins identified with EY, Y, and HEY as the carrier proteome as well as without any carrier proteome (Fig. 1b). The carrier proteomes primarily dictated which proteins were identified in different samples. Moreover, this pronounced bias correlated directly with the carrier levels. The results suggest that carrier proteomes need to be properly weighed to act as impartial carriers for all proteins in SCCs. We next examined the total numbers of proteins identified and quantified at different carrier levels. As expected, including a CPC increased the numbers of identified proteins consistently with rising carrier levels (Fig. 1c, Supplementary Fig. 1). However, the number of human proteins with precise quantification across the 14 SCCs (CV ≤ 20%) peaked at relatively low carrier levels (42×). In the sample containing the highest carrier level (434×), the majority of identified proteins could not be reproducibly quantified (Fig. 1c).

**Impurities and a limited number of ions lead to worse quantification**. We explored the relationship between quantitative precision and the number of fragment ions, and found very high carrier levels led to worse correlations and inferior quantification performance despite a higher number of total ions accumulated for MS/MS scans (Supplementary Fig. 2). Two major factors are causing worse quantification at higher carrier levels: (1) impurities from the carrier channel; (2) a limited number of reporter ions. We compared averaged SNR of the 14 single-cell reporters (Av14) or a subset of 12 reporters with the least isotope impurities (Av12) with the respective CV values in the 14 or 12 SCCs for human peptide spectral matches (PSMs) and proteins (Fig. 1d, Supplementary Fig. 2, Supplementary Fig. 3). The CV values displayed negative correlations with the average SNR at both PSM and protein levels in agreement with the findings in other studies[5]. Higher carrier ratios limited the maximum single-cell SNR, as the signal of both reporter ions and peptide fragment ions primarily derived from the CPC (Fig. 1d). It should be noted that the reporter ion intensities, which are normalized by injection time, correlated worse with the CV values than SNR (Supplementary Fig. 2, Supplementary Fig. 3).

Furthermore, we observed a significant contribution of isotopic impurities, particularly from carrier channel 126. We calculated protein CV's in all 14 SCCs with either raw or impurity-corrected SNR of reporter ions and found impurity correction substantially increased the number of proteins with CV ≤ 20% in samples with carrier levels higher than 98x (Fig. 1e). At increased ratios, we observed that channel 126 produced noteworthy isotopic impurities in addition to those in the empty 127 C channel (Supplementary Note 2), which affected channel 128 C (126 + 2x13C), and importantly, 127 N (126 + 15 N). Of note, Schoof et al.[9] and Cheung et al.[5] also noticed this negative impact of channel 127 N but Cheung et al. ascribed it to ion coalescence. The impurities explain worse quantification at high carrier ratios if ignored. In fact, TMT can accurately quantify ratios higher than 400 even for channel 127 N after impurity correction (Fig. 1f). Unfortunately, impurity correction also led to higher variations of quantified ratios and the correction for 15 N is not available in most data processing tools (Supplementary Note 3). Due to the negative impact by 127 N and 128 C, we calculated the CV of all PSMs and proteins without these two channels, resulting in much more accurate and reproducible quantifications on the 12 unaffected channels (Fig. 1d, e, Supplementary Fig. 3). Next, we aimed to evaluate the overall quantitative accuracy across all 14 SCCs in yeast peptides by comparing their relative intensities against the expected values (Supplementary Fig. 4). Similar to quantitative precision, accuracy was highly dependent on SNR. Distributions of relative intensities in SCCs were highly dispersed at low SNR and they converged to expected values with increased SNR. As the Av14 values in samples with high carrier levels (210x and 434x) were limited, the abundance ratios were distributed almost randomly and led to poor quantification accuracy.

**Impact of the CPC for detecting significantly regulated proteins**. To assess the impact of the CPC for detecting significantly regulated proteins, we took advantage of the known protein ratios in our mixed-species samples and examined the sensitivity and specificity of the CPC approach. We utilized the predefined relative species abundances between channels of a ratio of 2 for yeast peptides, 0.5 for E. coli peptides, and 1 for human peptides (Supplementary Note 4). We used the four ratio estimates to perform a t-test (represented by a volcano plot) analysis to identify significantly regulated PSMs with log2-fold change higher than 0.5 at $p < 0.05$. In almost all cases, less than 1% of human PSMs were wrongly assigned as significantly regulated, suggesting a high specificity (Fig. 2a, b). Despite the lower number of PSMs identified, samples without any carrier were most likely to assign the highest percentage of identified yeast and E. coli peptides as correctly regulated; however, this sensitivity in identifying significantly regulated peptides decreased as carrier levels increased. Ultimately, the highest number of regulated peptides were

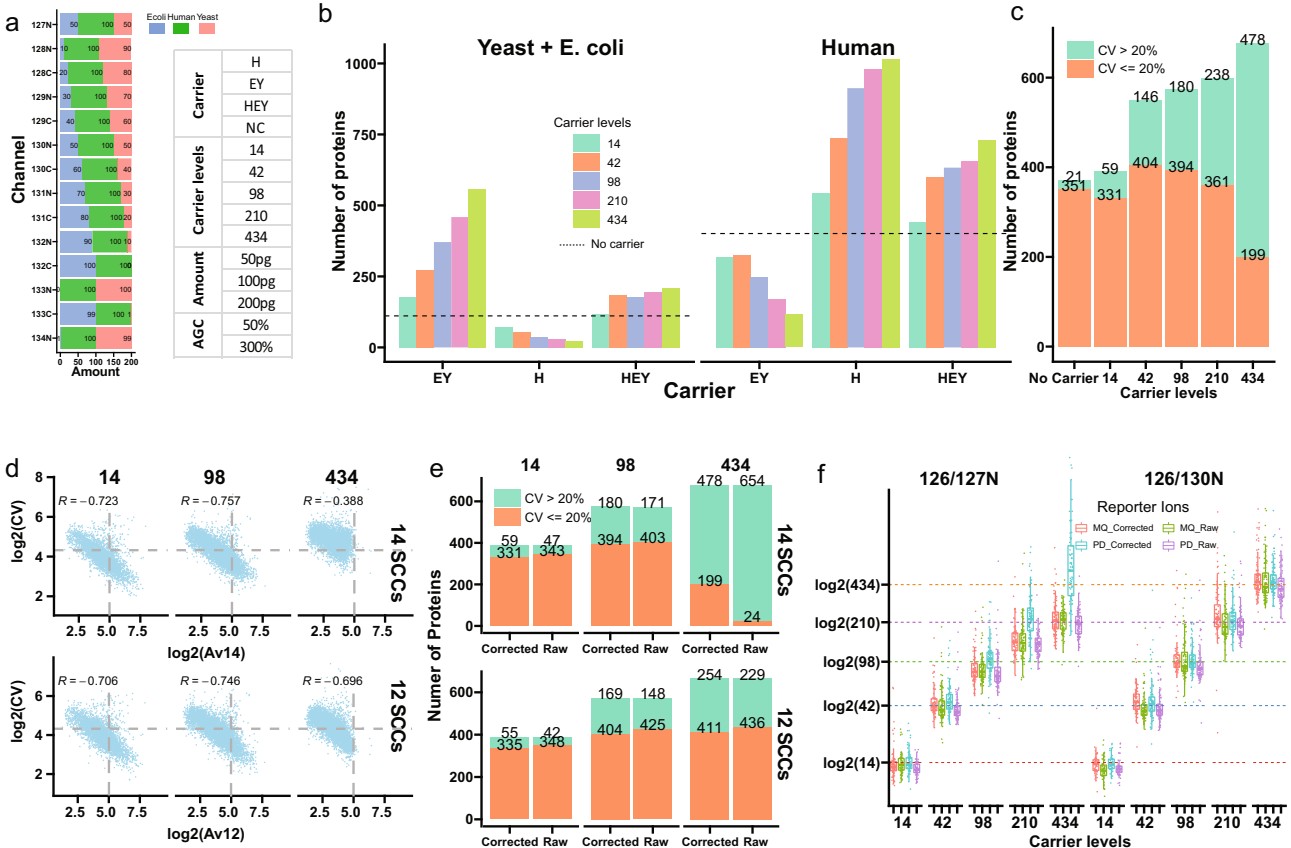

**Fig. 1 Explore the carrier effect with TMTpro in mixed-species samples. a** Depiction of TMTpro-labeled mixed-species samples and mass spectrometry acquisition settings. Digested peptide mixtures from yeast, E.coli and human HeLa cells were labeled with TMTpro in 14 SCCs and mixed in different proportions. **b** Number of identified proteins from different organisms. Samples with the following settings were selected: no carrier and carrier proteome as HEY, H, and EY; AGC300%; 50 pg per SCC; replicate 2. **c** Number of identified proteins with different carrier channel ratios. Samples with the following settings were selected: no carrier and carrier proteome; AGC300%; 50 pg per SCC; replicate 2. **d** Relationship between average reporter ion SNR of 14 SCCs (upper panel) or 12 SCCs (lower panel) and CV in identified PSMs from human proteins. Samples with the following settings were selected: carrier proteome as HEY; carrier levels at 14 and 434; AGC300%; 50 pg per single-cell channel. PSMs with Av14 $\geq$ 1 were used. Spearman's rank correlation coefficient was used as a measure of rank correlation. **e** Number of identified proteins with CV > 20% and CV $\leq$ 20% using either raw or corrected SNR values. CV values were calculated with either 14 SCCs (upper panel) or 12 SCCs (lower panel). Samples with the following settings were selected: carrier proteome as HEY; carrier levels at 14 and 434; AGC300%; 50 pg per single-cell channel; replicate 2. **f** Distribution of $\log_2(126/130\,N)$ and $\log_2(126/127\,N)$ in all human PSMs using reporter ion values from different methods. Methods for reporter ion values included raw and impurity-corrected values from MaxQuant and Proteome Discoverer. Samples with the following settings were selected: carrier proteome as HEY; AGC300%; 200 pg per single-cell channel.

detected in samples with 98x carrier. Conversely, only a small percentage of yeast and E. coli peptides were accurately quantified in samples with very high carrier levels (210× and 424×) despite the highest numbers of identified peptides. To determine whether the remaining regulated peptides were among the high abundant proteins, we plotted the log2-fold changes against the SNR values of channel 133 N, which represents the highest abundance of yeast peptides (Fig. 2c). At all carrier levels, the regulated peptides revealed higher SNR values than non-regulated peptides. Nevertheless, due to the suppression from the carrier proteome, overall abundances of yeast peptides at SCC were limited and many relatively high abundant peptides were not calculated as significantly regulated cases at very high carrier levels.

**Boost the SNR with NCE and MS/MS resolution**. We tested the most direct MS parameters, the normalized collisional energy (NCE) and MS/MS resolution to enhance SNR for quantification accuracy (Supplementary Note 1). We found elevated NCE levels (35–38%) at lower MS/MS resolution were the best compromise between quantification accuracy and identification. In accordance

with a previous study[8], NCE levels between 32 and 35% gave most PSMs (Fig. 3a). However, numbers of PSMs with CV $\leq$ 20% generally increased as NCE was correspondingly increased, particularly at high carrier levels. This was due to consistent increases of reporter SNR with higher NCE (Fig. 3b), despite the Sequest HT score function XCorr and MaxQuant Andromeda[10] scores peaking at lower NCE (Supplementary Fig. 5). We simultaneously observed a steady decrease of single-cell SNR values as carrier levels increased (Fig. 3b) indicating that even NCE level at 38% could not overcome the limits caused by the CPC. Higher MS/MS resolution resulted in a higher fraction of identifications with CV $\leq$ 20%; however, this came at the cost of a significantly reduced number of PSMs (Fig. 3c, d) due to the slow scan speed. To be noted, most of the current TMT-based SCP applications work with very high ion injection times, allowing very long transient times of the Orbitrap measurement without introducing additional overhead. Meanwhile, we envision that the ion injection times will be significantly reduced with the technical development in sample preparation, new MS instrumentation and the need for high-throughput analysis in SCP.

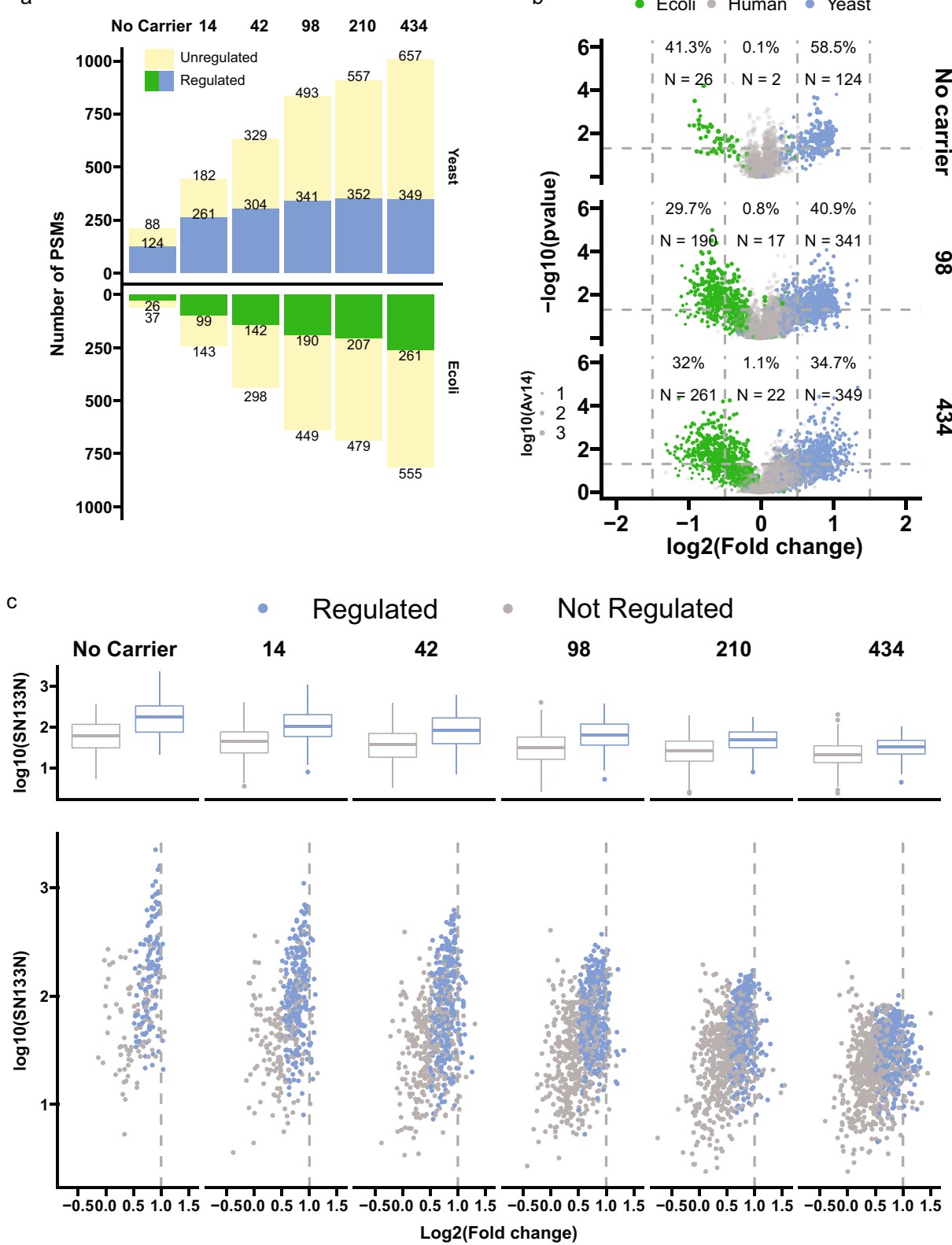

**Fig. 2 Impact of the carrier proteome for detecting significantly regulated proteins. a** Numbers of PSMs accurately quantified with selected TMT channels. From the 14 SCCs, we calculated log$_2$ ratios including, 131 C/132 N, 131 C/132 N, 131 C/132 N, and 131 C/132 N for yeast and human PSMs, 128 N/128 C, 129 N/130 C, 129 C/131 C, and 130 N/132 C for E.coli PSMs. PSMs with Parent intensity fraction ≥ 0.98 and Av14 ≥ 1 were kept. Samples with the following settings were selected: carrier proteome not as H; AGC300%. **b** Volcano plots of number of PSMs accurately quantified with selected TMT channels. N number of PSMs. The dash lines indicate cutoffs in fold change and p-values. **c** Abundance distribution (upper panel: boxplot; lower panel: scatter plot) of regulated and non-regulated yeast peptides. The Y-axis displays SN values of channel 133 N which represents the highest abundance of yeast peptides. In both **b** and **c** calculations of numbers were the same as **a**. PSMs with log2-fold change higher than 0.5 at $p < 0.05$ were designated as regulated.

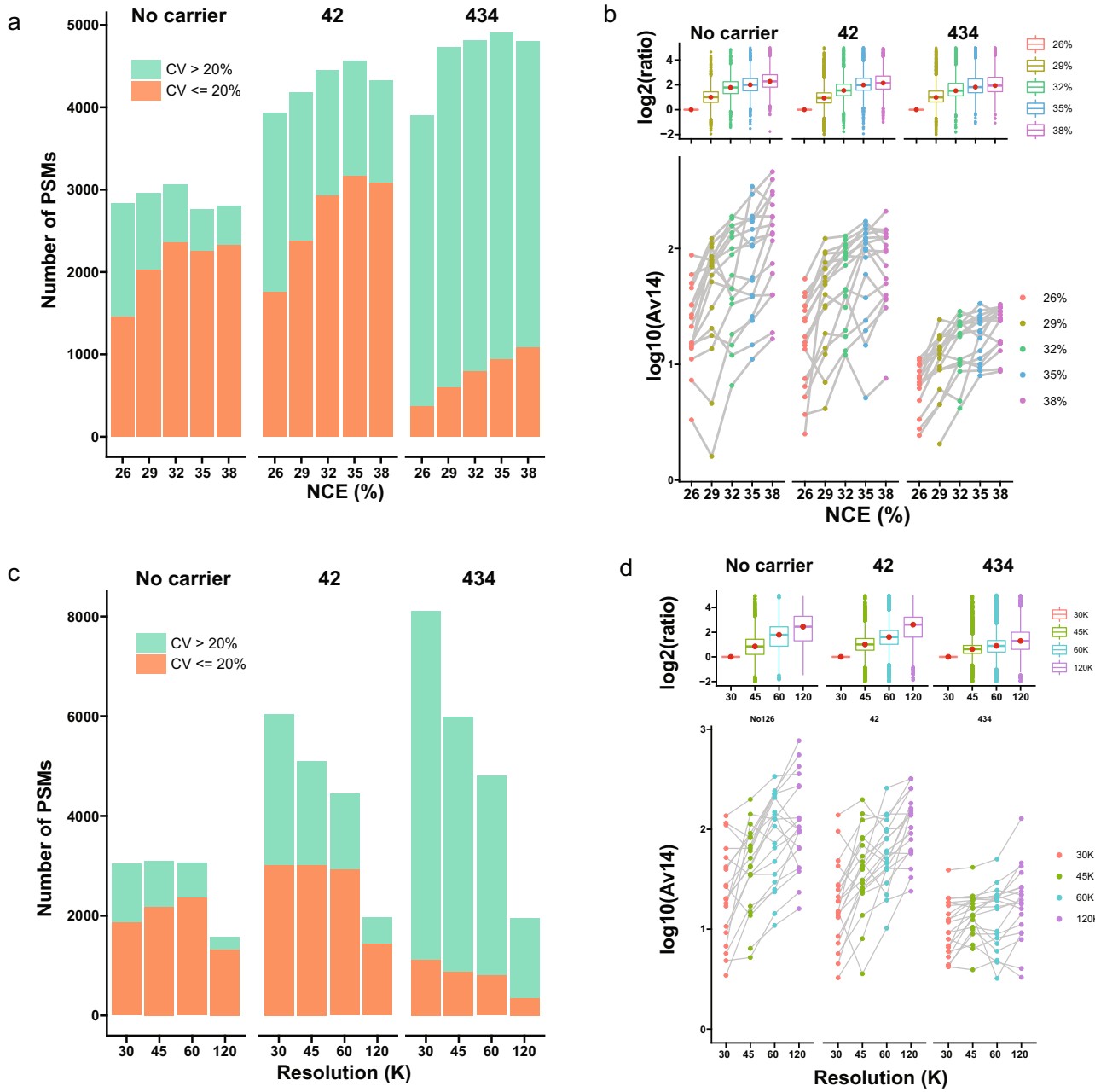

**Fig. 3 Increased normalized collisional energy and MS/MS resolution enhance SNR for better quantification accuracy. a** Number of identified PSMs with CV > 20% and CV ≤ 20% with different normalized collisional energies (NCE). **b** Distribution of Av14 in scans with different NCEs from randomly selected identical precursors (lower panel); boxplot of the $\log_2$ ratios of Av14 at different NCEs divided by Av14 at NCE 26%. In both **a** and **b**, samples with the following settings were selected: no carrier and carrier proteome as HEY; no carrier and carrier levels at 14 and 434; NCE at 26%, 29%, 32%, 35%, and 38%. Resolution was set to 60 K in all samples. **c** Number of identified PSMs with CV > 20% and CV ≤ 20% with different resolutions. **d** Distribution of Av14 in scans with different resolutions from randomly selected identical precursors (lower panel); boxplot of the $\log_2$ ratios of Av14 at different resolutions divided by Av14 at 30 K. In both **c** and **d** Samples with the following settings were selected: no carrier and carrier proteome as HEY; no carrier and carrier levels at 14 and 434; Resolution at 30, 45, 60, and 120 K. NCE was set to 32% in all samples.

**Intensity and SNR result in different global protein expressions**. For profiling cellular heterogeneity based on global protein expressions[11] with the isobaric carrier approach, protein abundances from reporter ions are first extracted and then subjected to dimensionality reduction methods, such as principal component analysis (PCA). To estimate relative protein copy numbers in proteomes, the intensity-based absolute quantification (iBAQ)[12] is the method of choice. Therefore, it is essential to evaluate the accuracy of protein abundances derived from reporter ions. We compared 4 different reporter ion abundance values (SNR and

intensities both as raw and impurity corrected, Supplementary Note 5) and demonstrated that they resulted in different protein abundance estimates especially in samples with low AGC target (Supplementary Fig. 6). Since the protein copy number estimates in our SCP model should match between the CPC and SCCs for each species, we tested the correlations between iBAQ values computed from full scans (MS1) with the CPC and SCCs (Fig. 4). Protein abundances at MS1 were calculated as summed abundances of identified peptides, where the Minora algorithm in Proteome Discoverer was used to perform untargeted feature

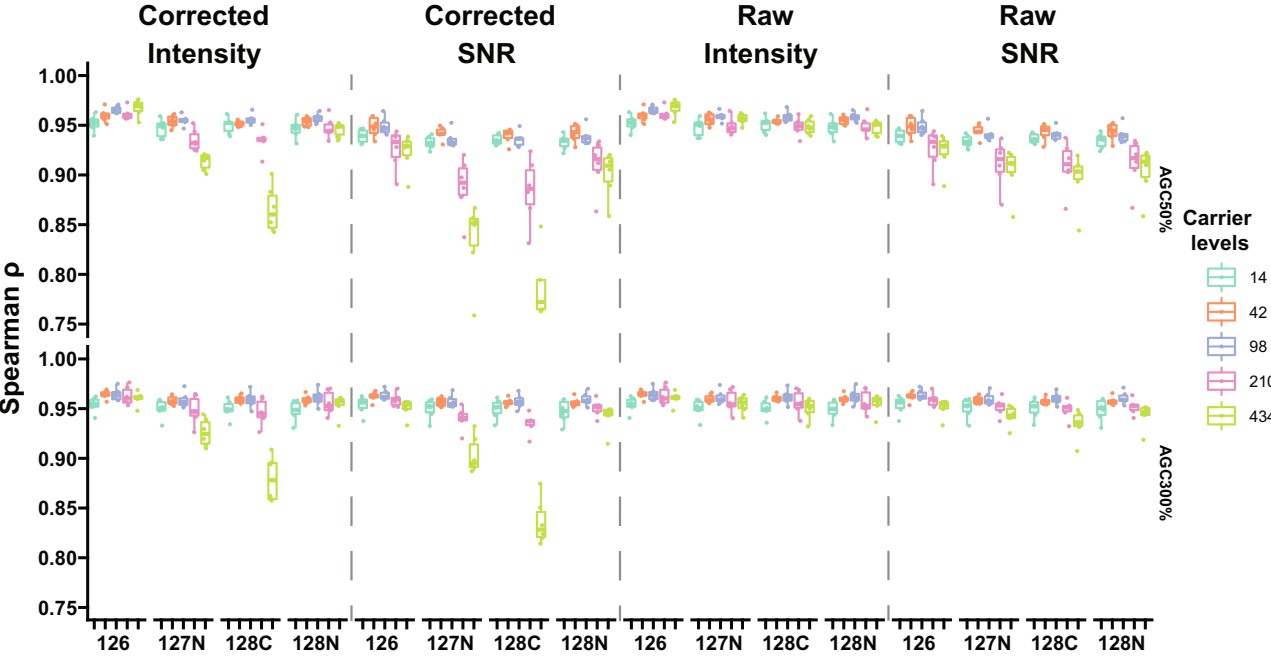

**Fig. 4 Correlation between MS1 abundances and abundances from selected reporter ion channels.** MS1 abundances were calculated from Minora node in Proteome Discoverer. Four different kinds of reporter ion abundances were calculated including raw and impurity-corrected intensities, raw, and impurity-corrected SNR. The Spearman's rank correlation coefficient was used as a measure of rank correlation.

**Table 1 Recommended settings in Orbitrap instruments for isobaric labeling-based single-cell proteomics.**

| Parameter | Recommended setting | Rationale |
|---|---|---|
| AGC | ≥AGC300% | Higher AGC target allows more ions in MS2 scans |
| NCE | 35% | Slightly higher NCE leads to higher reporter ion SNR without reducing peptide fragment quality |
| Resolution in MS2 | ≥60 K | • To resolve isobaric reporter ions |
|  |  | • To make use of the long fill time needed to reach the high AGC target |
| Carrier levels | 127 N and 128 C included: ≤100× | Impurities from TMTpro126 are substantial with very high carrier levels |
|  | 127 N and 128 C excluded: >100× |  |

detection for the peptides. Protein abundances on reporter ions were calculated as summed quantities of identified peptides from reporter ion abundances. Unlike the quantification of a single protein across TMT channels, the intensity values correlated better with MS1 abundances than SNR values, especially with low AGC settings (Supplementary Note 6). This is likely due to the fact that both reporter ion intensity values and MS1 abundances are scaled based on injection times. Furthermore, carrier levels of 42× and 98× showed the best correlations in almost all settings. For a TMT-based single-cell proteomics analysis in a biologically-relevant setting, our data suggested that researchers should notice the difference between the SNR and intensity values. Ideally, both of them should be extracted but used for different purposes.

## Discussion

In conclusion, our study systematically explored the effects of the isobaric carrier approach using a defined mixed-species model and provides a guideline for future SCP experiments (Table 1). Our finding that the carrier proteome specifically boosts the identification of the proteins contained within it opens the door for a variety of "targeted" SCP experiments. As we studied the tradeoff between identifications and quantitation at large carrier levels (>100×), we observed that the underlying reasons were a compression of the dynamic range of single-cell SNR at high

carrier levels, and for specific channels the impurities from the carrier channel. Therefore, we suggest excluding channels 127 N and 128 C for SCP experiments with extreme carrier levels. We tested the sensitivity and specificity of identifying significantly regulated proteins and our model suggests an optimal carrier level of ~100× when analyzing 14 SCCs. We recommend using reporter ion SNR for fold-change-based quantifications across channels and reporter intensities for protein copy number estimation within each channel. A higher NCE of up to 35% achieves better quantification performance by enhancing reporter abundances while maintaining peptide identifications.

Some other important aspects of SCP should also be explored and discussed in the future. For instance, the quality of the MS data in real SCP experiments is generally lower than bulk cell analysis as they suffer from elevated noise and background signals, which cannot be directly filtered out with current instrumentations. It would also be of interest to test how the carrier selection and elevated levels impact single-cell clustering based on protein abundances in a real biological setting. Other parameters and MS settings, such as Stepped Collision Energy and Real-Time Search, may have important roles in the field. In the future, the performance of SCP will benefit from the development of isobaric tags with higher multiplexing capacity (18-plex[13]), more sensitive instrumentation, and a higher dynamic range of mass analyzers. This study provides a roadmap to benchmarking such new developments.

## Methods

**Sample preparation**. Human epithelial cervix carcinoma HeLa cells were cultured in DMEM (Gibco, Invitrogen) as previously described[14]. Cells were harvested at ~80% confluence after washing twice with PBS (Gibco, Life technologies). E.coli were grown on LB medium plates and colonies were scrapped manually and transfered to 1.5 ml tubes. E. coli were resuspended in PBS buffer and washed 3 times followed by centrifugation to pellet the cells and discard the supernatant. For HeLa and E.coli cells, boiling 4% SDS in 50 mM Tris pH 8.5 was added to the cells. The tube was heated for 10 min at 95 degrees, and DNA/RNA was sheared by sonication with a tip. Tryptophan assay was utilized to determine protein concentration followed by reduction and alkylation with TCEP and CAA. Sample prep was performed using protein aggregation capture[15] during which proteins were aggregated onto magnetic beads and digested overnight sequentially with Lys-C (1:200 protease to protein ratio) for 2 h at 37 C and Trypsin (1:50) overnight. Mass spec-compatible yeast intact (undigested) extracts were brought from Promega (Catalog number: V7341) and processed according to the technical manual. All the digest supernatant was cleaned using C18 solid-phase extraction and the peptide concentration was determined using nano-drop. Digested peptides were labeled with TMTpro following the manufacturer's protocol. TMTpro-labeled peptides from different species were pooled with different ratios as described.

**LC-MS/MS**. All samples were analyzed on an Orbitrap Exploris 480 mass spectrometer coupled with the Evosep One system using an in-house packed capillary column with the pre-programmed 30 samples-per-day gradients in data-dependent acquisition mode. The column temperature was maintained at 60 °C using an integrated column oven (PRSO-V1, Sonation, Biberach, Germany). Spray voltage was set to 2 kV, funnel RF level at 40, and heated capillary temperature at 275 °C. Full MS resolutions were set to 120,000 at $m/z$ 200 and the full MS AGC target was 300% with an IT of 25 ms. Mass range was set to 350–1400. The intensity threshold was kept at 1E5. Isolation width was set at 0.8 $m/z$. All data were acquired in profile mode using positive polarity. AGC target value, resolution and normalized collision energy (NCE) were set differently for individual samples. Maximum injection time was set to Auto.

**Data processing and analysis**. All raw files were processed in Proteome Discoverer 2.4 (Thermo Fisher Scientific) and MaxQuant with the human, yeast and E.coli Uniprot Reference Proteome database without isoforms (January 2019 release). Trypsin was set as the digesting enzyme and up to one missed cleavage was allowed. TMTpro was specified as a fixed modification on lysine and peptide N-terminus, carbamidomethylation of cysteine was specified as fixed modification and methionine oxidation was specified as a variable modification. Precursor and fragment mass tolerances were set to 10ppm and 0.02 Da in Sequest HT, respectively. Specifically, reporter abundance was based on either SNR or intensity, both with raw and impurity-corrected values.

**Statistics and reproducibility**. No normalization or scaling was applied. Protein grouping was done with the default settings in Protein Grouping and Protein FDR Validator Node in Proteome Discoverer. Reporter quantification was done with the default settings in Reporter Ions Quantifier Node: Co-Isolation Threshold was set to 50; Protein Roll-Up (protein abundance) was set to Use All Peptides. All the files were processed in batch mode to get result files individually. A modified modification.xml file was used in MaxQuant to enable TMTpro based database search. All the statistical analysis was conducted with in-house written R-scripts using default settings. For all boxplots[16], the lower and upper hinges correspond to the first and third quartiles. The upper whisker extends from the hinge to the largest value no further than 1.5 × IQR from the hinge (where IQR is the inter-quartile range). The lower whisker extends from the hinge to the smallest value at most 1.5 × IQR of the hinge. Data beyond the end of the whiskers are called "outlying" points.

**Reporting summary**. Further information on research design is available in the Nature Research Reporting Summary linked to this article.

## Data availability

The mass spectrometry proteomics data have been deposited to the ProteomeXchange Consortium via the PRIDE[17] partner repository with the dataset identifier PXD027742. All source data underlying the graphs and charts can be found in Supplementary Data 1. A full list of sample descriptions, MS settings, identified proteins, and PSMs can be found in Supplementary Data 2. Any remaining information can be obtained from the corresponding author upon reasonable request.

## Code availability

The in-house written R-scripts for the data analysis are publicly available at https://github.com/ZiluYe/TMT_Carrier_Analysis.

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

## Acknowledgements

This work was supported by Novo Nordisk Foundation (NNF14CC0001 and NNF17SA0027704), and the program of excellence from the University of Copenhagen (CDO2016). This work has also been supported by EPIC-XS, project number 823839, funded by the Horizon 2020 program of the European Union.

## Author contributions

Z.Y., T.B., and J.V.O. conceived and designed the study; Z.Y. and T.B. contributed with experimental data; Z.Y., T.B., P.R., and J.V.O. contributed with data interpretations; Z.Y. wrote the manuscript; and all authors edited and approved the final version.

## Competing interests

The authors declare no competing interests.
