## [Transparent Peer Review File · Communications Biology]

This manuscript has been previously reviewed at another Nature Portfolio journal. This document only contains reviewer comments and rebuttal letters for versions considered at Communications Biology.

Reviewers' comments:

Reviewer #1 (Remarks to the Author):

The revised version of the manuscript has addressed some of my initial concerns. Specifically, the new figure format allows for better readability and the requested additional information is now provided within the manuscript. I agree with Reviewer 1 and have already expressed my concerns previously, that while the authors have addressed some issues raised by the reviewers the novelty of their findings remains questionable. However, the scope of communications biology might be a better fit to the proposed manuscript.

Additional comments:

The emphasis to differentiate between S/N and intensity quantification as suggested by both reviewers and Cheung et al is highly appreciated. I nevertheless would like to see more elaboration on this topic:

- It would be of interest if the regulation displayed in Figure 2b is influenced by either intensity or S/N based quantification.

- Do the remaining regulated yeast proteins in the high-carrier samples represent the most abundant proteins - in detail, are just low abundant precursor lost due to signal suppression by the carrier? This would be of great value for the general scp community.

- To quality control RI signals in carrier experiments SCPCompanion was designed by Cheung et al., would such S/N filtering of low S/N PSMs improve accuracy in the described fold change evaluation? (Fig. 2a)

Figure legend line 252 the old figure design is still referenced

Reviewer #2 (Remarks to the Author):

I would like to thank the authors for addressing our concerns and adapting their text accordingly. The manuscript has now been transferred to another journal. The authors have, in my opinion, addressed most of my concerns bar the following:

Regarding Query / Response 3:

Thank you for the clarification. This topic is indeed not discussed in great detail in the literature; however this work is also not the first time it has been observed and described. The authors should at least acknowledge and cite the observation that 128C and 127N are affected, as described in e.g. Schoof et al., Supplementary Fig. 2.

Regarding Query / Response 4:

I think the quantification accuracy is a very important aspect to further investigate, and the impact of the work submitted by Ye et al. would be significantly enhanced through further clarification of the underlying reasons. I would hypothesize that the ratio compression/reduced dynamic range observed in Figure 2 is solely a result of lower S/N values in the single-cell channels, and could be mitigated by higher ion injection times as shown in both Cheung et al., Fig 4 and Schoof et al., Fig 3f. One explanation would in that case be that the increased carrier level is not the reason for the ratio compression, but the lower S/N values in the single-cell channels due to AGC target enforcing lower injection times. Why I want to stress this point is due to the fact that the experimental results observed here (using 60k, 118ms only which is vastly shorter than the majority of "real" single cell proteomics experiments in the literature), would likely be different when a higher AGC target would be used to enforce the maximum IT for all MS2 spectra or in general higher ion injection times would be used. Thus, the authors have probably identified the optimal carrier level (98x) for their specific ion injection time and AGC target and peptide amount in the single-cell channels, but this might not translate to a true single cell setting. As a result, changing one of these parameters could result in a different recommendation for carrier levels. The manuscript would be significantly strengthened

through improved data analysis and interpretations in this regard.

Minor comment:

Supplementary Table 1 is a 1.5Gb spreadsheet, which might not be the most efficient way of communicating the MS run settings etc. The authors should consider providing MS run methods separately from the actual identification data.

Point-by-point rebuttal letter:

COMMSBIO-21-3155-T:

"A deeper look at carrier proteome effects for single-cell proteomics" by Zilu Ye et al.

We have carefully gone through all of the comments made by the reviewers and addressed them point-by-point below. Our answers to the reviewer questions are indicated in blue text. We would like to thank the reviewers for their input and suggestions to the manuscript text and figures, which we have updated accordingly. This helped us improve our manuscript during this process.

Reviewer Comments:

Reviewer #1 (Remarks to the Author):

The revised version of the manuscript has addressed some of my initial concerns. Specifically, the new figure format allows for better readability and the requested additional information is now provided within the manuscript. I agree with Reviewer 1 and have already expressed my concerns previously, that while the authors have addressed some issues raised by the reviewers the novelty of their findings remains questionable. However, the scope of communications biology might be a better fit to the proposed manuscript.

Additional comments:

The emphasis to differentiate between S/N and intensity quantification as suggested by both reviewers and Cheung et al is highly appreciated. I nevertheless would like to see more elaboration on this topic:

Query 1: It would be of interest if the regulation displayed in Figure 2b is influenced by either intensity or S/N based quantification.

Response 1: As mentioned in the manuscript, the regulated numbers in Figure 2 were calculated at PSM level. And at PSM level, SN and intensity values have almost no difference on the ratios of different channels. As shown in the Figure 1 below, the numbers of regulated PSMs were almost the same with reporter ion values from either SN or intensity. The minor difference came from the PSMs with very low SN values.

Figure 1 | Impact of different reporter ion values for detecting significantly regulated proteins.

Meanwhile, the SN and intensity values do have effects on the quantification at protein level. Figure 2 (corresponding to Figure 1c in the manuscript) shows the numbers of proteins with CV cutoffs using different reporter ion values. At the most extreme carrier level (434x), SN showed better quantification performance than intensity.

Figure 2 | Numbers of proteins with CV cutoffs using different reporter ion values.

Since this effect is not significant and to avoid complexity, we decide not to include this analysis in the manuscript.

Query 2: Do the remaining regulated yeast proteins in the high-carrier samples represent the most abundant proteins - in detail, are just low abundant precursor lost due to signal suppression by the carrier? This would be of great value for the general scp community.

Response 2: We agree it is a good idea and could be of great value. Hence, we made a new figure (Figure 2c) showing the correlation between the fold changes of the yeast PSMs and the SN value of channel 133N which represents the highest abundance of yeast peptides.

We also added the following sentences in the manuscript:

“To determine whether the remaining regulated peptides were among the high abundant proteins, we plotted the \log_2 -fold changes against the SN values of channel 133N, which represents the highest abundance of yeast peptides (Fig. 2c). At all carrier levels, the regulated peptides revealed higher SN values than non-regulated peptides. Nevertheless, due to the suppression from the carrier proteome, overall abundances of yeast peptides at SCC were limited and many relatively high abundant peptides were not calculated as significantly regulated cases at very high carrier levels.”

Query 3: To quality control RI signals in carrier experiments SCPCompanion was designed by Cheung et al., would such S/N filtering of low S/N PSMs improve accuracy in the described fold change evaluation? (Fig. 2a)

Response 3: We have tried the tool with our datasets. Nevertheless, it gives different SN values than ours, which were extracted from Proteome Discoverer. Since we have showed extensively the relationship between quantification and SN values in our manuscript and especially the newly addressed **Query&Response 2** with the new Figure 2c, we do not think it is still necessary to show such analysis in our manuscript.

Figure legend line 252 the old figure design is still referenced

Changed, thank you!

Reviewer #2 (Remarks to the Author):

I would like to thank the authors for addressing our concerns and adapting their text accordingly. The manuscript has now been transferred to another journal. The authors have, in my opinion, addressed most of my concerns bar the following:

Regarding Query / Response 3:

Thank you for the clarification. This topic is indeed not discussed in great detail in the literature; however this work is also not the first time it has been observed and described. The authors should at least acknowledge and cite the observation that 128C and 127N are affected, as described in e.g. Schoof et al., Supplementary Fig. 2.

Response: Thank you, we now cited the Schoof *et al.* in the manuscript.

Regarding Query / Response 4:

I think the quantification accuracy is a very important aspect to further investigate, and the impact of the work submitted by Ye et al. would be significantly enhanced through further clarification of the underlying reasons. I would hypothesize that the ratio compression/reduced dynamic range observed in Figure 2 is solely a result of lower S/N values in the single-cell channels, and could be mitigated by higher ion injection times as shown in both Cheung et al., Fig 4 and Schoof et al., Fig 3f. One explanation would in that case be that the increased carrier level is not the reason for the ratio compression, but the lower S/N values in the single-cell channels due to AGC target enforcing lower injection times. Why I want to stress this point is due to the fact that the experimental results observed here (using 60k, 118ms only which is vastly shorter than the majority of “real” single cell proteomics experiments in the literature), would likely be different when a higher AGC target would be used to enforce the maximum IT for all MS2 spectra or in general higher ion injection times would be used. Thus, the authors have probably identified the optimal carrier level (98x) for their specific ion injection time and AGC target and peptide amount in the single-cell channels, but this might not translate to a true single cell setting. As a result, changing one of these parameters could result in a different recommendation for carrier levels. The manuscript would be significantly strengthened through improved data analysis and interpretations in this regard.

Response: We agree with the hypothesis. While we could further increase the AGC target to evaluate this effect, we did compare two different AGC targets (50% and 300%) in our analysis (**Supplementary Figure 2 & 3, Supplementary Note 6**). As shown in Supplementary Figure 3a (right-bottom panel), increased AGC target settings indeed resulted in higher S/N values for the single-cell channels. The reason we did not discuss this topic extensively in our manuscript was simply because it has been discussed before in other SCP studies and we would like to avoid repetition. In terms of ion injection times, maximum injection time was set to Auto and the highest injection time we have was 247ms, which is close to the typical setting used in previous SCP studies. Nevertheless, we also discussed this topic in our last revision:

“To be noted, most of the current TMT-based SCP applications work with very high ion injection times, allowing very long transient times of the Orbitrap measurement without introducing additional overhead. Meanwhile, we envision that the ion injection times will be significantly reduced with the technical development in sample preparation, new MS instrumentation and the need of high-throughput analysis in SCP.”

Minor comment:

Supplementary Table 1 is a 1.5Gb spreadsheet, which might not be the most efficient way of communicating the MS run settings etc. The authors should consider providing MS run methods separately from the actual identification data.

Response: Thanks and we agree. We have made a new excel table with only the run method section and insert the link to get full access to all identifications in the table.

REVIEWERS' COMMENTS:

Reviewer #1 (Remarks to the Author):

After a very intensive review process, the authors of this manuscript were able to explain all concerns in a point-by-point summary. Therefore, I support the publication of this manuscript.

Reviewer #2 (Remarks to the Author):

The authors have in their latest revision addressed my concerns with the work appropriately.

Point-by-point rebuttal letter:

COMMSBIO-21-3155-T:

"A deeper look at carrier proteome effects for single-cell proteomics" by Zilu Ye et al.

We would like to thank the reviewers for their input and suggestions during the review process. This helped us improve our manuscript.

Reviewer Comments:

Reviewer #1 (Remarks to the Author):

After a very intensive review process, the authors of this manuscript were able to explain all concerns in a point-by-point summary. Therefore, I support the publication of this manuscript.

Reviewer #2 (Remarks to the Author):

The authors have in their latest revision addressed my concerns with the work appropriately.